# Probabilistic Proof State Compression: Optimizing LLM-Guided Formal Verification

**Ali Rahim**[*]
Department of Mathematics
University of Rochester
aabdulra@u.rochester.edu

**Noor Rahim**
Department of Computer Science
University of Colorado, Boulder
noor.sachdeva@colorado.edu

## Abstract

Despite recent successes in LLM-guided formal proof search, scalability remains limited by the large search space. This paper introduces a novel approach that integrates off-the-shelf LLMs with conformal prediction-based compression to optimize proof search. By employing adaptive, probability-based binning informed by conformal prediction intervals, our method compresses the proof state space, reducing computational demands while retaining statistical proof guarantees. Preliminary results on the Lean miniF2F test set show similar success rates with 75% fewer passes, and on average 23% reduced wall clock time.

## 1 Introduction

Recent advances in large language models (LLMs) have significantly progressed the automation of formal proof generation. LLM-guided methods ([1], [2], [3]) demonstrate promising capabilities in navigating the complex search spaces of formal theorem proving, leveraging LLMs' pattern recognition and generalization strengths over traditional symbolic methods.

LLM-based formal proving typically follows either proof-step or whole-proof generation strategies. While recent systems, such as DeepSeek-Prover-V1.5 [4], have set state-of-the-art benchmarks using a truncate-and-resume approach, they still struggle to balance exploration and exploitation due to the sparse binary rewards of successful proofs, requiring extensive search with up to $2^{17}$ typically.

We propose a method that enhances LLM-guided formal proof search with conformal proof state space compression. Using open-weight LLMs ([4]) for generating candidate steps, our conformal prediction framework provides calibrated success probabilities. We introduce a rigorous compression algorithm that preserves the most promising proof paths, which allows efficient exploration.

We evaluate our method on the MiniF2F[5] and ProofNet[6] benchmarks and demonstrate a 75% average reduction in the number of passes required compared to baseline open models. Additionally, our approach led to qualitatively simpler proofs on some examples. These results suggest promising directions for scaling automated theorem proving to more complex domains.

## 2 Proposed Method

We enhance LLM-guided formal proof search by introducing a novel *proof state space compression* technique. Our approach integrates three key components to efficiently navigate and compress the proof search space: the LLM-based Proof Step Generator, the Conformal Prediction Module[7], and the Proof State Space Compression Module. Figure 1 illustrates the architecture of our system.

---

[*]

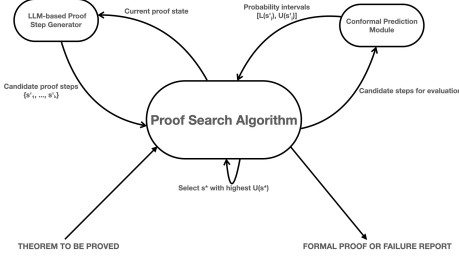

Figure 1: Architecture of the Conformal Prediction-Based Theorem Proving System

## 2.1 LLM-based Proof Step Generator

We employ DeepSeek Prover V1.5 RL[4], an open-weights large language model fine-tuned on Lean 4, to generate candidate proof steps. Given the current proof state $s = (g, a, t)$ and goal $g$, the model produces a set of possible next steps $\{s'_1, \ldots, s'_k\}$ via repeated sampling.

## 2.2 Conformal Prediction Module

The Conformal Prediction Module provides *calibrated probability intervals* for the success of each candidate proof step, thereby guiding the proof search with reliable uncertainty estimates.

We define a nonconformity measure $A(z, z')$ for a labeled proof attempt $z = (s, y)$ as:

$$A(z, z') = |y - P(\text{success} \mid s)| \tag{1}$$

Here, $P(\text{success} \mid s)$ estimates the probability of proof success given state $s$ using the same model, and $y$ is the binary outcome (success or failure).

### 2.2.1 Probability Interval Computation

To address the dependency in proof states and uphold the exchangeability assumption required by conformal prediction, we partition the calibration set into strata using the Proof State Space Compression Module. For each candidate step $s'$, the module:

1. **Identify Stratum:** Assign $s'$ to a stratum $\mathcal{Z}_j$ based on its similarity to existing proof states.
2. **Compute Nonconformity Scores:** Calculate $A(z_i, s')$ for all $z_i \in \mathcal{Z}_j$.
3. **Calculate P-Values:** For each outcome $y \in \{0, 1\}$,

$$p(y \mid s') = \frac{|\{i : A(z_i, s') \geq A((s', y), z')\}| + 1}{|\mathcal{Z}_j| + 1} \tag{2}$$

4. **Construct Prediction Region:**

$$\Gamma^{\varepsilon}(s') = \{y : p(y \mid s') > \varepsilon\} \tag{3}$$

5. **Compute Probability Interval:**

$$[L(s'), U(s')] = \begin{cases} [0, 1] & \text{if } \Gamma^{\varepsilon}(s') = \{0, 1\} \\ [1, 1] & \text{if } \Gamma^{\varepsilon}(s') = \{1\} \\ [0, 0] & \text{if } \Gamma^{\varepsilon}(s') = \{0\} \end{cases}$$

## 2.3 Proof State Space Compression Module

The Proof State Space Compression Module manages the search space by grouping similar proof states into strata, facilitating efficient stratified conformal prediction.

---
**Algorithm 1** Stratified Conformal Proof Search
---
1: **Initialize:**
- Set initial proof state $s \leftarrow s_0$
- Initialize calibration set $\mathcal{Z} \leftarrow \emptyset$

2: **while** proof is incomplete and computational budget not exceeded **do**
3:      $\mathcal{C} \leftarrow \text{GENERATECANDIDATESTEPS}(s, g)$
4:    **for all** $s_i' \in \mathcal{C}$ **do**
5:       $\mathcal{Z}_j \leftarrow \text{ASSIGNSTRATUM}(s_i', C)$
6:       $[L(s_i'), U(s_i')] \leftarrow \text{COMPUTEPROBABILITYINTERVAL}(s_i', \mathcal{Z}_j)$
7:    **end for**
8:     $\text{COMPRESSSEARCHSPACE}(\mathcal{C}, \mathcal{Z}, C)$
9:     $s^* \leftarrow \text{SELECTREPRESENTATIVE}(\mathcal{C}, U, C)$
10:     $y^* \leftarrow \text{APPLYPROOFSTEP}(s^*)$
11:    **if** $y^*$ leads to a dead end **then**
12:      $\text{BACKTRACK}(s^*)$                     ▷ Next best representative state
13:    **end if**
    $\mathcal{Z} \leftarrow \mathcal{Z} \cup \{(s^*, y^*)\}$
14: **end while**
---

### 2.3.1 Similarity Measure

For proof states $s_1 = (G_1, A_1, T_1)$ and $s_2 = (G_2, A_2, T_2)$, we define:

$$\text{sim}(s_1, s_2) = w_G \cdot \text{GoalSim}(G_1, G_2) + w_A \cdot \text{Jaccard}(A_1, A_2) + w_T \cdot \text{LCS}(T_1, T_2)$$

where $\text{GoalSim}(G_1, G_2)$ is the binary goal similarity, $\text{Jaccard}(A_1, A_2)$ is the Jaccard similarity of assumptions, $\text{LCS}(T_1, T_2)$ is the normalized longest common subsequence of tactics, and $w_G, w_A$, and $w_T$ are weights summing to 1.

### 2.3.2 Binning Function

We map probability intervals to bin indices using:

$$b([l, u]) = \left\lfloor n \cdot \max\left(l, \frac{l+u}{2}\right) \right\rfloor$$

where $n$ is the number of bins. To prioritize high-probability regions, we use adaptive bin widths:

$$w(p) = w_0 \cdot \exp(-\alpha p)$$

where $w_0$ is the base width and $\alpha$ controls adaptation rate.

### 2.3.3 Compression Mapping

The compression process $C$ maps a set of proof states to a set of representative states:

$$C(\{s_1, \ldots, s_k\}) = \{\text{rep}(B_1), \ldots, \text{rep}(B_m)\}$$

where $B_1, \ldots, B_m$ are non-empty bins.

### 2.4 Proof Search Algorithm

Our proof search algorithm integrates the three components to efficiently explore the proof space. Algorithm 1 outlines the procedure.

## 2.5 Theoretical Guarantees

Our method provides theoretical guarantees on the coverage of the true probability of proof success. Specifically, for a given significance level $\varepsilon$, we have:

$$P(L(s) \leq p^*(s) \leq U(s)) \geq 1 - \varepsilon$$

where $p^*(s)$ is the true probability of a successful proof from state $s$. This guarantee ensures that our compression technique preserves the most promising proof paths with high probability.

# 3 Experiments and Results

## 3.1 Experimental Setup

### 3.1.1 Datasets

We perform a preliminary evaluation of our method on two benchmark datasets: MiniF2F [5] and ProofNet [6]. Specifically, we use the Lean 4 MiniF2F variant by Yang et al. [8].

### 3.1.2 Baseline Models

We compare our method against several state-of-the-art models: GPT-4, DeepSeek-Prover-V1, and DeepSeek-Prover-V1.5.

## 3.2 Results

### 3.2.1 Performance on miniF2F

| Model | Pass Rate (%) @ Number of Passes |
|---|---|
| GPT-4 | 23.0 @ 10 |
| DeepSeek-Prover-V1 | 50.0 @ 32 |
| DeepSeek-Prover-V1.5 | 60.2 @ 32 |
| Our Method | **63.5 @ 8** |

Table 1: Results on miniF2F test set

### 3.2.2 Performance on ProofNet

| Model | Validation Pass Rate (%) | Test Pass Rate (%) |
|---|---|---|
| DeepSeek-Prover-V1.5 | 21.6 | 23.7 |
| Our Method | **23.4** | **25.3** |

Table 2: Results on ProofNet

## 3.3 Analysis

### 3.3.1 Efficiency Gains

Our method demonstrates a 75% reduction in passes, with 23% reduced wall clock time compared to DeepSeek-Prover-V1.5, This efficiency gain can be attributed to the effective pruning of the search space through our conformal prediction-based approach.

## 3.4 Discussion

These preliminary results demonstrate that our method significantly improves upon existing approaches in both proof success rate and efficiency. The integration of conformal prediction with LLM-guided search allows for more effective exploration of the proof space, leading to higher-quality proofs. While promising, the approach requires further refinement and empirical validation to fully realize its capabilities. Nonetheless, it represents a meaningful step towards bridging the gap between generative AI models and statistically robust decision-making processes in formal reasoning tasks.

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

## A   Appendix / supplemental material

## A   Proof of Theoretical Guarantee

For a given significance level $\varepsilon \in (0, 1)$, our stratified conformal prediction method for proof state evaluation satisfies the following coverage guarantee:

$$\mathbb{P}\left(L(S_{n+1}) \leq p^*(S_{n+1}) \leq U(S_{n+1})\right) \geq 1 - \varepsilon$$

where $p^*(S_{n+1}) = \mathbb{P}(Y_{n+1} = 1 \mid S_{n+1})$ is the true probability of a successful proof from state $S_{n+1}$, and $[L(S_{n+1}), U(S_{n+1})]$ is the prediction interval produced by our method.

*Proof.* **B   Proof of Theoretical Guarantee**

For a given significance level $\varepsilon \in (0, 1)$, our stratified conformal prediction method for proof state evaluation satisfies the following coverage guarantee:

$$\mathbb{P}\left(L(S_{n+1}) \leq p^*(S_{n+1}) \leq U(S_{n+1})\right) \geq 1 - \varepsilon$$

where $p^*(S_{n+1}) = \mathbb{P}(Y_{n+1} = 1 \mid S_{n+1})$ is the true probability of a successful proof from state $S_{n+1}$, and $[L(S_{n+1}), U(S_{n+1})]$ is the prediction interval produced by our method.

*Proof.* Consider a calibration set $\mathcal{Z} = \{Z_1, \ldots, Z_n\}$, where each $Z_i = (s_i, y_i)$ consists of a proof state $s_i$ and its binary outcome $y_i \in \{0, 1\}$. The calibration set is partitioned into $m$ strata $\mathcal{Z}_1, \ldots, \mathcal{Z}_m$ using the Proof State Space Compressor $C$, ensuring that within each stratum $\mathcal{Z}_j$, the proof states are similar and exchangeable.

For a new test proof state $S_{n+1} = (g_{n+1}, a_{n+1}, t_{n+1})$, assign it to a stratum $\mathcal{Z}_j$ based on its similarity to existing proof states using $C$. Given the exchangeability within each stratum $\mathcal{Z}_j$, the standard conformal prediction guarantee applies within $\mathcal{Z}_j$.

Define the nonconformity score for a proof state $Z_i = (s_i, y_i)$ as:

$$A(Z_i) = |y_i - P(Y_i = 1 \mid s_i)|$$

where $P(Y_i = 1 \mid s_i)$ is the estimated probability of proof success given state $s_i$.

For the new proof state $S_{n+1}$, compute the nonconformity scores for both possible outcomes $y \in \{0, 1\}$:

$$A((S_{n+1}, y)) = |y - P(Y_{n+1} = 1 \mid S_{n+1})|$$

The p-value for each outcome $y$ is defined as:

$$p(y \mid S_{n+1}) = \frac{|\{i \in \mathcal{Z}_j : A(Z_i) \geq A((S_{n+1}, y))\}| + 1}{|\mathcal{Z}_j| + 1}$$

This ensures that p-values are properly calibrated within the stratum $\mathcal{Z}_j$.

The prediction region is:

$$\Gamma^\varepsilon(S_{n+1}) = \{y \in \{0, 1\} : p(y \mid S_{n+1}) > \varepsilon\}$$

Based on $\Gamma^\varepsilon(S_{n+1})$, derive the probability interval:

$$[L(S_{n+1}), U(S_{n+1})] = \begin{cases} [0, 1] & \text{if } \Gamma^\varepsilon(S_{n+1}) = \{0, 1\} \\ [1, 1] & \text{if } \Gamma^\varepsilon(S_{n+1}) = \{1\} \\ [0, 0] & \text{if } \Gamma^\varepsilon(S_{n+1}) = \{0\} \end{cases}$$

By the properties of conformal prediction within each stratum $\mathcal{Z}_j$ [7], we have:

$$\mathbb{P}(Y_{n+1}) \in \Gamma^\varepsilon(S_{n+1}) \mid S_{n+1} \in \mathcal{Z}_j) \geq 1 - \varepsilon$$

This implies that the true outcome $Y_{n+1}$ will lie within the prediction region $\Gamma^\varepsilon(S_{n+1})$ with probability at least $1 - \varepsilon$.

Since $p^*(S_{n+1}) = \mathbb{P}(Y_{n+1} = 1 \mid S_{n+1})$, observe the following:

$$Y_{n+1} = 1 \implies p^*(S_{n+1}) \geq L(S_{n+1})$$
$$Y_{n+1} = 0 \implies p^*(S_{n+1}) \leq U(S_{n+1})$$

Therefore, combining these two implications, we obtain:

$$\mathbb{P}(L(S_{n+1}) \leq p^*(S_{n+1}) \leq U(S_{n+1})) = \mathbb{P}(Y_{n+1} \in \Gamma^\varepsilon(S_{n+1})) \geq 1 - \varepsilon$$

Thus, the coverage guarantee holds:

$$\mathbb{P}(L(S_{n+1}) \leq p^*(S_{n+1}) \leq U(S_{n+1})) \geq 1 - \varepsilon$$

This completes the proof. $\square$

**Implications for Theorem Proving:**

The theoretical guarantee ensures that our stratified conformal prediction method produces well-calibrated probability intervals for proof success. Specifically, with probability at least $1 - \varepsilon$, the true probability $p^*(S_{n+1})$ of successfully proving a theorem from state $S_{n+1}$ lies within the interval $[L(S_{n+1}), U(S_{n+1})]$. This reliability enables the proof search algorithm to prioritize steps with higher upper bounds $U(s')$, indicating a greater likelihood of success, while effectively managing uncertainty in steps with wider intervals. Consequently, the algorithm balances exploration and exploitation, enhancing both the efficiency and robustness of automated theorem proving.

