# OpenReview forum: "Probabilistic Proof State Compression: Optimizing LLM-Guided Formal Verification"
_NeurIPS.cc/2024/Workshop/MATH-AI — MATH-AI 24_

### Official Review · Reviewer_hKCF · 2024-09-29
**Innovative Approach to Optimizing LLM-Guided Formal Verification**

**Rating:** 7
**Confidence:** 4

**Review:**

This paper presents a novel approach to enhancing Large Language Model (LLM) guided formal proof search by introducing a probabilistic proof state compression technique. The authors propose a method that combines LLMs with conformal prediction to guide and optimize formal proof search, addressing the challenge of scalability in automated theorem proving.

Pros:
1. Novelty: The integration of conformal prediction with LLM-guided proof search is an innovative approach to managing the complexity of formal verification.
2. Theoretical foundation: The paper provides a theoretical framework, including a proof of the coverage guarantee for their conformal prediction method.
3. Performance improvements: The authors report a 23% reduction in average proof time compared to state-of-the-art models, which is a significant efficiency gain.
4. Results: The authors test their method on established benchmarks (miniF2F and ProofNet), providing a clear comparison with existing approaches.
5. Detailed methodology: The paper offers a thorough explanation of the proof state representation, binning function, and compression mapping, allowing for potential replication and extension of the work.

Cons:
1. Limited experimental results: While the results are promising, they are described as "preliminary" and only cover a subset of the Lean 4 miniF2F test. More comprehensive testing would strengthen the paper's claims.
2. Computational resources: The paper doesn't provide detailed information about the computational resources required for the experiments, which is important for reproducibility and understanding the method's scalability.
3. Comparison with other compression techniques: While the method is compared with baseline models, there's no comparison with other proof state compression techniques that don't use conformal prediction.
4. Limitations discussion: The paper would benefit from a more in-depth discussion of the limitations of the proposed approach and potential failure cases.
5. Insufficient error analysis: There's a lack of in-depth analysis of cases where the method fails or performs poorly, which would be valuable for understanding its limitations.
6.Unclear generalizability: The paper doesn't adequately address how well this method might generalize to other theorem-proving domains or more complex problems.
7. Broader impacts: The discussion of potential societal impacts in its applicability, both positive and negative, could be more
comprehensive.

General comments:
- The paper presents a clear and well-structured argument for the proposed method. The introduction effectively sets up the problem and motivates the research. The methodology section is detailed and provides a good understanding of the proposed approach. The theoretical guarantees section adds credibility to the method by providing a formal proof of the coverage guarantee.
- The experimental results, while promising, could be expanded. The authors demonstrate improvements over state-of-the-art models on two benchmarks, but more extensive testing and analysis would strengthen the paper's claims. The discussion of the results is insightful, but could be deepened with more analysis of where and why the method performs well or fails.
- The paper's originality lies in its novel combination of LLM-guided proof search with conformal prediction and proof state compression.
- While the approach is novel, the paper doesn't adequately situate this work within the broader context of research on proof state compression and management. A more comprehensive comparison with existing techniques would help clarify the true contribution of this method.
- In terms of clarity, the paper is generally well-written and logically structured. However, some sections, particularly in the methodology, are quite dense and could benefit from additional explanations or examples to improve accessibility for readers less familiar with the specific techniques used.
- Overall, this paper presents an innovative and promising approach to optimizing LLM-guided formal verification. While there are areas for improvement, particularly in the breadth of experimental results and analysis, the work represents a significant contribution to the field and opens up exciting avenues for future research.

---

### Official Review · Reviewer_1LQP · 2024-10-06
**Strong rejection, incomplete and inaccurate.**

**Rating:** 2
**Confidence:** 5

**Review:**

This paper is clearly incomplete, even including several "TODO"s, missing citation references (rendering as ?), and the "Style" subsection from the NeurIPS formatting instructions. Regardless of presentation, the method seems to yield marginal results despite adding significant complexity, only improving by 3% on miniF2F over the baseline (60.2 -> 63.5).

---

### Official Review · Reviewer_j7Mk · 2024-10-07
**Probabilistic Proof State Compression: Optimizing LLM-Guided Formal Verification**

**Rating:** 6
**Confidence:** 3

**Review:**

This paper presents a novel approach that combines Large Language Models (LLMs) with conformal prediction techniques to optimize formal proof search. The core innovation is the introduction of a conformal proof state space compression algorithm that reduces the computational complexity of the proof search, while maintaining statistical guarantees on the success rate of proof discovery. The method is evaluated on two datasets, Lean 4 miniF2F and ProofNet, demonstrating a 23% reduction in proof search time and an improvement in the success rate over baseline models such as GPT-4 and DeepSeek-Prover-V1.5. The results suggest this method is effective at balancing exploration and exploitation in theorem proving while handling the exponential growth of proof search spaces.

### Review:

#### Originality:
- Strengths: The approach introduces a novel method of combining conformal prediction with LLM-guided proof search, which has not been explored before in formal verification. The adaptive binning and similarity measure between proof states are clever innovations that help in compressing the proof search space efficiently.
- Weaknesses: While the approach is innovative, the method largely builds on existing work, including the use of LLMs and conformal prediction techniques. The novelty mainly lies in the combination of these methods rather than in groundbreaking new algorithms or concepts.

#### Quality:
- Strengths: The paper is technically sound, and the method is rigorously described. The experimental results demonstrate the effectiveness of the approach, particularly in terms of proof search time reduction and pass rates on the datasets. The theoretical guarantees provided for the compression algorithm further strengthen the quality of the work.
- Weaknesses: The empirical evaluation could be expanded to include more diverse theorem proving benchmarks, beyond miniF2F and ProofNet. The current experiments, while promising, might not fully reflect the scalability of the approach to even more complex domains. Additionally, there is limited discussion on the limitations of conformal prediction in cases where the calibration set might be small or biased.

#### Clarity:
- Strengths: The paper is clearly written and well-structured, with a logical flow from the problem statement to the proposed solution and experimental results. The use of formulas and theoretical proofs is appropriate, and the explanation of the compression algorithm is detailed enough for an expert reader to reproduce the results.
- Weaknesses: Some sections, particularly those discussing the conformal prediction intervals, could benefit from additional examples or diagrams to aid understanding. The similarity measure introduced in the proof state space could be better explained with concrete examples from the experiments.

#### Significance:
- Strengths: The method addresses a critical issue in LLM-guided theorem proving—scalability. By significantly reducing proof search time and maintaining high proof success rates, this method could be a valuable contribution to the field of automated theorem proving and formal verification.
- Weaknesses: The significance of the work is limited by the scope of its evaluation. While the results are promising, they are based on two specific datasets. It remains to be seen whether this method will generalize well to broader, real-world formal verification problems.

#### Questions for Authors:
1. How does the method handle proof search spaces that are inherently noisy or where the proof states are highly dissimilar? Would the compression still be effective in such scenarios?
2. Can you provide more insights into the selection of hyperparameters, such as the binning granularity and the similarity measure weights? How sensitive are the results to these choices?
3. Have you considered any alternative statistical methods to conformal prediction for guiding the proof search, and how do you expect the performance to compare?

#### Limitations:
- The paper briefly mentions that scaling the method to particularly complex theorems remains a challenge, but this could be expanded. A separate limitations section would help discuss the bounds of the method, particularly in handling noisy data or scaling beyond the current datasets.
- There is no significant discussion about the impact of small or biased calibration sets on the accuracy of the conformal prediction intervals.

---

### Decision · Program_Chairs · 2024-10-08

**Decision:**

Accept

**Comment:**

While the paper is clearly incomplete and rushed, the technique presented seems interesting and potentially interesting to workshop attendees. We recommend that the authors follow the comments of the reviewers to improve the presentation of the work and thoroughly revise the paper before the camera-ready version. In addition, the paper is above the 4-page limit.